

# Improving the age constraints on the archeological record in Scladina Cave (Belgium): new speleothem U-Th ages and paleoclimatological data.

Hubert B. Vonhof[1], Sophie Verheyden[2], Dominique Bonjean[3], Stéphane Pirson[4,5], Michael Weber[6], Denis Scholz[6], John Hellstrom[7], Hai Cheng[8,9], Xue Jia[8], Kevin Di Modica[3], Gregory Abrams[10,3], Marjan A.P. van Nunen[11], Joost Ruiter[11], Michèlle van der Does[12], Daniel Böhl[6], and H. Jeroen L. van der Lubbe[11]

[1]Max Planck Institute for Chemistry, Mainz, Germany
[2]Royal Belgian Institute of Natural Sciences, Brussels, Belgium
[3]Scladina Cave Archaeological Centre, Espace muséal d'Andenne, Andenne, Belgium
[4]Wallonia Heritage Agency, Scientific and Technical Direction, Namur, Belgium
[5]University of Liège, Department of Geology and European Archaeometry Centre, Liège, Belgium
[6]Johannes Gutenberg University, Institute for Geosciences, Mainz, Germany
[7]University of Melbourne, Melbourne, Australia
[8]Institute of Global Environmental Change, Xi'an Jiaotong University, Xi'an 710054, China
[9]Institute of Earth Environment, Chinese Academy of Sciences, Xi'an 710061, China
[10]ArcheOS – Research Laboratory for Biological Anthropology, Department of Archaeology, Ghent University, Ghent, Belgium
[11]Vrije Universiteit Amsterdam, Institute of Earth Sciences, Amsterdam, the Netherlands
[12]Alfred Wegener Institute, Helmholtz Centre for Polar and Marine Research, Bremerhaven, Germany.

*Correspondence to*: Hubert B. Vonhof (Hubert.vonhof@mpic.de)

**Abstract.** The sedimentary sequence in Scladina Cave (Belgium) is well-known for its rich Middle Paleolithic assemblages
and its numerous faunal remains. Of particular interest is the presence of a nearly complete mandible of a Neandertal child. To place all these finds in the correct chronostratigraphic context, various dating techniques have been applied over the past decades. This resulted in a reasonably well-constrained age model, roughly spanning the last glacial cycle. Age constraints of the lower part of the Scladina sequence as well as from the underlying Sous-Saint-Paul Cave were however absent until now. Previous attempts to date several speleothem layers in Scladina Cave, using U-Th dating were only partly successful,
presumably because diagenetic alteration of speleothem material compromised the ages. In the present study we re-assessed U-Th dating of various speleothem levels in Scladina Cave, applying state-of-the-art U-Th dating, and carefully selecting material that experienced little to no diagenetic alteration. The new results constitute a robust age framework for the Scladina sequence, which provides precisely dated stratigraphic anchor points that improve the previous age model. Furthermore, new U-Th analyses, for speleothems from the lower part of the Scladina sequence and from the Sous-Saint-Paul sequence, document
Middle Pleistocene ages, making this one of the longer fossil-rich cave sedimentary sequences in NW Europe. The new data





confirm that speleothem deposition predominantly took place in periods of warmer climate, while siliciclastic sediments characterize the colder intervals. New speleothem ages further suggest that the Neandertal mandible found in the sequence, and previously placed in Marine Isotope Stage 5a or 5b, could potentially be as old as Marine Isotope Stage 5d.

## 1    Introduction


Scladina Cave is situated near the Belgian town Andenne in a valley branching off the Meuse River (Fig. 1). The cave has been investigated since 1971, following the discovery of Middle Palaeolithic artefacts at the entrance of the cave (Bonjean et al., 2014). The cave at that time was nearly completely filled with a stack of siliciclastic sediments and intercalated flowstone levels. After several decades of careful scientific excavation, a significant part of the sedimentary infill has been removed (Fig.

2), and the cave continues to deliver a wealth of archaeological and paleontological information (Bonjean et al., 2014).

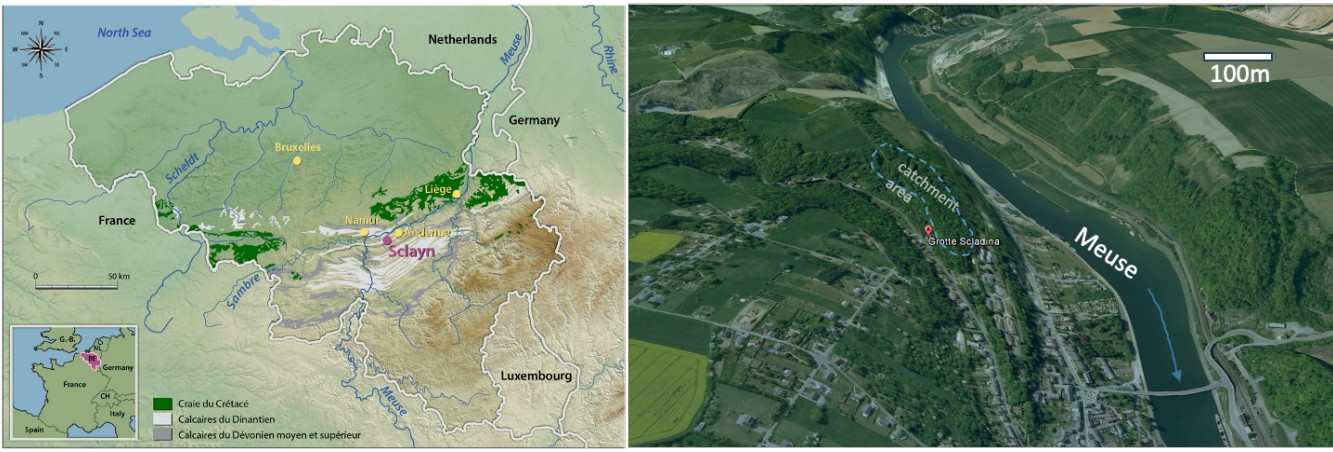

**Figure 1: Left: Map of Belgium with the town Sclayn indicated, where Scladina Cave is situated. Right: Aerial view of Scladina Cave, accessible from the Pontine valley, (© Google Earth 2020). Indicated in blue is our rough approximation of the catchment**
**area of dripwater in the cave.**

Arguably the most significant fossils retrieved from the cave are the mandible, a fragment of right maxillary and sixteen teeth of a juvenile (Bonjean et al., 1996; Otte et al., 1993; Toussaint and Bonjean, 2014) Owing to careful excavation methods and favourable conditions for the preservation of ancient DNA in Scladina Cave, it has been possible to extract valuable genomic

information from the mandible (Orlando et al., 2006; Peyregne et al., 2019). Besides this Neandertal material, a wealth of artefacts (stone tools) and fossil faunal remains have been retrieved from the sedimentological sequence in the cave (Bonjean





et al., 2014; Otte, 1992), and geochemical analyses on that material have provided valuable insight in the paleoenvironmental and palaeoecological setting (Bocherens et al., 1997; Bocherens, 2014).

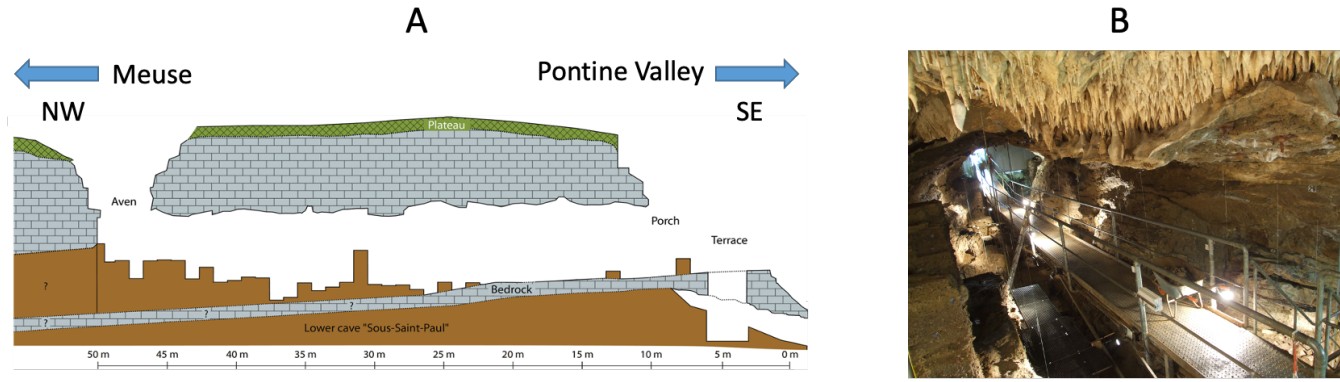

**Figure 2: A) Schematic cross section of Scladina Cave and the underlying Sous Sain Paul Cave (Bonjean et al., 2014). The entrance at the SE side is nowadays closed by a door. B) scaffolding and walkways installed in the cave, and positioning lines that mark the archeological excavation grid.**

The chronostratigraphic context of the finds from the cave has been reconstructed using a variety of dating techniques, anchored in meticulous reconstruction of the lithostratigraphy and sedimentary dynamics (Bonjean, 1998; Pirson, 2007, 2014; Pirson et al., 2014b; Pirson et al., 2008). The dating technique that is often most reliable in cave settings, Uranium series dating of speleothem CaCO$_3$, however, did not yield very precise ages initially (Bonjean, 1998; Gewelt et al., 1992). Presumably this was due to diagenetic alteration of speleothem calcite, as most speleothem material has been buried in siliciclastic sediments in Scladina Cave, and degradation of speleothem calcite is commonly observed in the excavated material.

Summarizing all the dating efforts done so far, a consistent age model is available for the spatially complex Scladina Cave infill stratigraphy (Pirson, 2014; Pirson et al., 2014b). In that age model, flowstone levels separating the different siliciclastic stratigraphic units generally seem to represent the warmer climate periods through the last glacial cycles, as is also the case in other caves in Belgium (Quinif, 2006). However, the accuracy of the existing age model for Scladina has not been easy to establish, as the dating methods used have significant uncertainties, some parts of the sedimentary infill are difficult to date and some discrepancies between age estimates remain (Pirson et al., 2014b). Furthermore, the age model of the lower part of the sequence, including the sediments in the underlying Sous-Saint-Paul Cave is essentially unknown.

In the present paper, we have reinvestigated the potential for Uranium series dating of speleothem material in Scladina Cave. This is done because significant progress has been made in U-series dating of speleothem material over the past decades (Cheng et al., 2013; Scholz and Hoffmann, 2008), and because new, and better-preserved speleothem material has been excavated from the cave. The objective of this study was to carefully select the best-preserved flowstone and stalagmite





material in the cave. In doing so, we targeted all known speleothem levels that were thick enough to preserve good quality speleothem calcite. We furthermore included two flowstone levels from the Sous-Saint-Paul Cave, the sediments of which

stratigraphically underlie Scladina Cave (Otte, 1992; Pirson et al., 2008).

In addition to our dating efforts, we used several geochemical techniques on the speleothem material to investigate the diagenetic state of the material, and to retrieve paleoclimatological information.

## 2 Materials and methods

### 2.1 Speleothem sample collection

The current stratigraphic profile of the sediment infill of Scladina Cave, including the sediments from the underlying Sous-Saint-Paul Cave, is shown in Fig. 3. It contains several levels with speleothem growth, the most prominent of which is the currently still active speleothems growing on top of unit H. From this level we collected 2 stalagmites and a column for

analysis. A single speleothem was collected from layer 1B-BK (in unit 1B-BRUN). Two more stalagmites were collected from another prominent speleothem growth interval in unit 4A, previously dated to marine isotope stage (MIS) 5: one is coming from an in situ speleothem (CC4 in unit 4A-IP), while the other one consists of two halves of a stalagmite reworked from CC4 in a gully (unit 4A-CHE). From a speleothem complex in stratigraphic unit 6B, we collected two stalagmites, and from two more speleothem levels in Sous-Saint-Paul Cave, covering the lowermost part of the composite sequence, we cut flowstone

fragments. Slabs prepared from the samples collected are shown in Fig. A1. All material is listed in Table 1.

| Sample | stratigraphic Unit | sample type |
|---|---|---|
| 2010#1 | H | stalagmite |
| 2010#2 | H | stalagmite |
| 2012#5 | H | column |
| 2014#206 | 1B-BRUN | stalagmite |
| 2012#1&2 | 4A-CHE | stalagmite |
| 2010#14 | 4A-IP | stalagmite |
| 2010#6 | 6B | stalagmite |
| 2014#618 | 6B | stalagmite |
| 2012#4 | XVII | flowstone |
| 2012#3 | XIX | flowstone |

**Table 1: The material collected for this study. 2012#1&2 are two pieces of the same stalagmite in stratigraphic continuity with 2012#1**
**being the basal part.**




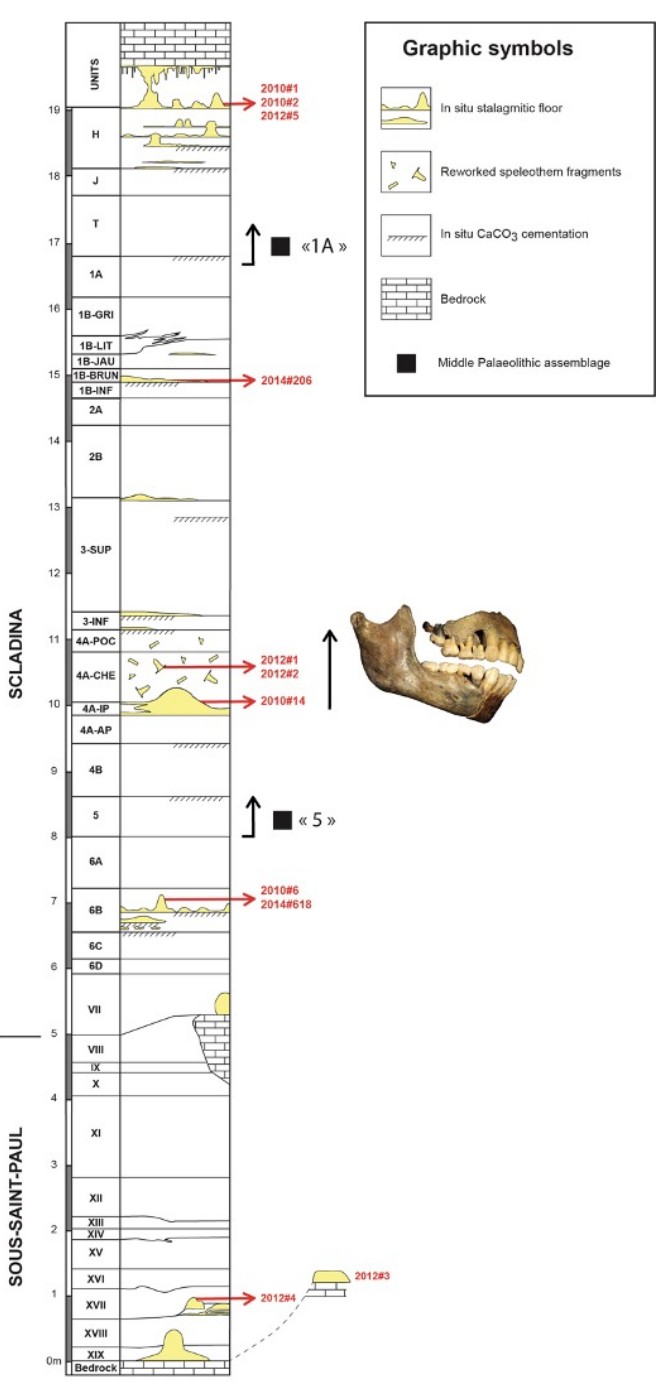

**Figure 3: Schematic stratigraphic column of the combined Scladina and Sous Saint Paul cave sedimentary sequence. Speleothem**
**occurrences are indicated in yellow coloring. Vertical scale bar is in meters. The position of the samples that were dated in this study is indicated by the red arrows. Photograph indicates the position of the Neandertal mandible and teeth found in unit 4A.**



## 2.2    U-series analysis

$^{230}$Th/U-dating of the selected material was performed over a period of several years in four different laboratories, as described below.

In Xi'An, China, samples were dissolved using $HNO_3$ and spiked with a solution of known $^{236}$U/$^{233}$U/$^{229}$Th ratio. U and Th were precipitated using a Fe solution and subsequently extracted using decontaminated resin columns and sequential washes of 6N HCl and ultra-clean water to extract Th and U, respectively. Finally, the solution was analysed on a Thermo Fisher

Neptune Plus multicollector inductively coupled plasma mass spectrometer (MC-ICP-MS). The entire procedure was carried out at the Institute of Global Environmental Change, Xi'an Jiaotong University (China) and followed established protocols (e.g. Cheng et al., 2020).

In Melbourne, Australia, sample preparation and chemical separation of U and Th and MC-ICP-MS measurements followed established techniques (Hellstrom, 2003). Samples were dissolved in concentrated nitric acid. A mixed enriched isotope tracer

($^{229}$Th–$^{233}$U–$^{236}$U) was added to the solution. The equilibrated solution was then loaded onto an ion exchange column containing Eichrom TRU and pre-filter resin to discard the calcite matrix and finally elute U and Th. The U–Th fractions were analysed on a Nu Instruments Plasma MC-ICP-MS at the University of Melbourne  (cf Weij et al., 2024).

At the Max Planck Institute for Chemistry (MPIC), Mainz Germany, subsamples of ca. 300 mg were extracted next to the growth axis of the speleothems, prepared by column chemistry and analysed using a Nu Plasma (Nu Instruments) MC-ICP-

MS. The weighed samples were briefly leached in 7N $HNO_3$ to remove potential surface contamination, dissolved in 7N $HNO_3$, and a mixed $^{229}$Th–$^{233}$U–$^{236}$U spike was added (see (Gibert et al., 2016), for details on spike calibration). Potential organic material was removed by adding a mixture of concentrated $HNO_3$, HCl and $H_2O_2$. The dried samples were then dissolved in 6N HCl, and U and Th were separated using ion exchange column chemistry (Yang et al., 2015). Details about the MC-ICP-MS procedures are described in (Obert et al., 2016).

At Johannes Gutenberg University (JGU), Mainz, Germany, sample preparation and analysis were performed at the Institute for Geosciences. The procedures were similar to those applied at MPIC. Samples were weighed, dissolved in 7N $HNO_3$ and subsequently spiked with a previously calibrated $^{229}$Th-$^{233}$U-$^{236}$U spike solution. The spike was gravimetrically prepared and then calibrated against IRMM074/10 (Richter et al., 2009) and a solution prepared from speleothem sample WM 1 from Wilder Mann Cave, which has an approximate age of 2.02 Ma (Meyer et al., 2009) and is therefore in secular equilibrium (Cheng et

al., 2013). The samples were evaporated to dryness and treated with concentrated $HNO_3$, HCl and $H_2O_2$, and then evaporated again, dissolved in 7N HNO3 and passed through ion exchange columns filled with 1.5 mL of Bio-Rad AG 1-X8 (200 – 400 mesh size) anion exchange resin to separate U and Th. The procedure was carried out twice and followed a method developed at the Berkeley Geochronology Center, which is based on the procedure described in (Edwards et al., 1987).



Mass spectrometric analyses of U and Th were conducted using a Thermo Fischer Scientific Neptune Plus MC-ICP-MS
equipped with an Elemental Scientific Apex Omega HF desolvator. U and Th were measured separately in a standard-bracketing procedure and the data were corrected offline using an in-house R-script (R Development Core Team, 2024) correcting for mass fractionation, ion counter yield, instrumental drift and tailing. Each sample was analysed in standard-sample-bracketing mode, accounting for instrumental drift. For U, the CRM112-A Uranium Isotopic Standard was analysed prior to and after each sample, while for Th, an in-house Th standard solution with known isotope ratios of $^{229}$Th, $^{230}$Th and $^{232}$Th was analysed. Mass fractionation was determined for each standard by analysing $^{235}$U/$^{238}$U or $^{229}$Th/$^{232}$Th, respectively. Within the same analysis, ion counter yield for IC1 was determined by analysing $^{234}$U/$^{238}$U or $^{230}$Th/$^{232}$Th, both previously corrected for mass fractionation. Tailing was determined externally by analysing a concentrated CRM 112-A U solution (containing $^{234}$U, $^{235}$U, $^{238}$U) or a natural Th solution (containing $^{232}$Th) at the beginning and the end of each analytical session. Abundance sensitivity was calculated and corrected for isotopes of $^{229}$Th, $^{230}$Th, $^{233}$U and $^{236}$U.

## 2.3    Calculation of ages and correction for detrital Th

All activity ratios and ages were calculated using the half-lives reported by Cheng et al. (2013) and corrected for detrital contamination assuming a $^{232}$Th/$^{238}$U weight ratio of 3.8 for the detritus and $^{230}$Th, $^{234}$U and $^{238}$U in secular equilibrium. To account for the uncertainty of the correction, we assumed an uncertainty of 50 %, which was fully propagated to the corrected activity ratios and ages. All uncertainties are reported at the 2σ-level.

## 2.4    Stable isotope analysis of CaCO₃

Samples were analysed on a Thermo Delta+ (at VUA Amsterdam) or a Delta-V mass spectrometer (at MPIC Mainz) both equipped with a Gasbench II preparation device. ~8-50 microgram of CaCO$_3$ sample, placed in a He-filled 12 ml exetainer vial, were digested in >99% H$_3$PO$_4$. Subsequently the CO$_2$-He gas mixture was transported to the Gasbench in 5.0-grade Helium carrier gas. In the Gasbench, water vapor and various gaseous compounds were separated from the He-CO$_2$ mixture prior to sending it to the mass spectrometer for isotope analysis, and reported as CaCO$_3$ δ$^{13}$C (δ$^{13}$C$_{cc}$) and CaCO$_3$ δ$^{18}$O (δ$^{18}$O$_{cc}$) on the "Vienna PeeDee Belemnite" (VPDB) scale.

The in-house CaCO$_3$ standards VICS (having a δ$^{13}$C value of 1.45‰ VPDB and a δ$^{18}$O value of -5.44‰ VPDB) was analyzed with each run of samples in both labs. CaCO$_3$ standard weights of standards are chosen so that they span the entire range of sample weights of the samples. After correction of isotope effects related to sample size the reproducibility of the standards typically is better than 0.1 ‰ (1SD) for δ$^{18}$O and 0.1 ‰ (1SD) for δ$^{13}$C.


## 2.5  Stable isotope analysis of dripwater

A single dripwater sample from Scladina Cave was analysed by use of a Thermo TC-EA pyrolysis furnace coupled to a Delta XP mass spectrometer at VUA Amsterdam. Analytical precision (1SD) of 4 in-house water standards (8 repetitive analyses

for each) in the sample series that contained this sample did not exceed 0.2 ‰ $\delta^{18}O$ and 1.0 ‰ for $\delta^{2}H$. Triplicate analysis of the sample itself resulted in a 1SD precision smaller than that of the standard water reproducibility. Isotope data of the dripwater sample is reported on the "Vienna Standard Mean Ocean Water" (VSMOW) scale.

## 2.6  Fluid inclusion isotope analysis


Fluid inclusion isotope analyses were performed on discrete ~ 0.5-1.0 g chips of speleothem calcite. These samples were crushed under a controlled atmosphere in a crushing cell attached to a Picarro L2140*i* water isotope analyser (De Graaf et al., 2020). The resulting hydrogen and oxygen isotope values of fluid inclusion water ($\delta^{2}H_{fi}$ and $\delta^{18}O_{fi}$ respectively) are reported on the VSMOW scale. Typical 1SD precisions for this technique, based on standard water injections in this system, are better

than 0.3 ‰ for $\delta^{18}O_{fi}$ values and 1.0 ‰ for $\delta^{2}H_{fi}$ values (De Graaf et al., 2020).

## 2.7  Laser ablation trace element analysis

Trace element concentrations were measured by laser ablation ICP-MS at the MPIC following the procedure of Jochum et al.

(2012), with a high-resolution sector-field ICP-MS ThermoElement2, interfaced with a New Wave UP-213 Nd:YAG laser ablation system. The silicate glass NIST SRM 612 was analysed for external calibration of the trace element analyses. Data are reported in  μg/g (ppm) units.

## 2.8  Cave site and regional climate


The mean annual surface temperature in the town of Andenne is ~10.4 degrees Celsius and average monthly temperatures typically range between 3 and 18 degrees Celsius (KMI, 2024).





Cave temperature monitoring from July 2010 until July 2011 showed an annual cycle between 8 and 13 degrees C, with an average annual value of 10.5 degrees C, which is dampened in seasonality, but very close to the mean annual surface temperature of the region. Drip rates in the cave have been observed to be variable, depending on rainfall amounts above the cave. This likely relates to the comparatively small hydrological catchment area of Scladina Cave. Relative humidity measurements in the cave taken at different points in time typically were close to 100%.

## 3    Results

### 3.1    Diagenetic screening of speleothem samples

The thinner speleothem layers commonly appear significantly altered. In the more prominent speleothem layers, particularly when developed in stalagmite morphology, the calcite generally appears visually better preserved. The effect of diagenetic alteration on speleothem material that was buried in siliciclastic sediments in the cave was geochemically characterized in a laser ablation ICP-MS transect measured at high spatial resolution from the visually well-preserved core of stalagmite 2014#618 (unit 6B) to its diagenetically altered outer surface (Fig. 4). The transect in total is 42 mm long.  Macroscopically, the diagenetic alteration is evident in whitish and yellowish discoloration and loss of visible lamination and transparency towards the more altered zone near the outer surface of the speleothem.

Clearly shown in the trace element data is that $^{238}$U and $^{232}$Th have high concentrations at the outer surface of the stalagmite, that has been in direct contact with the sediment in which it was buried. $^{238}$U shows a gradient of decreasing values towards the core of the speleothem, reaching stable ~0.1ppm concentrations in the central growth axis of the speleothem which represents the CaCO$_3$ material that appears unaltered at visual inspection. This gradient is much sharper for $^{232}$Th, which shows markedly increased values only at the outer surface of the stalagmite.

Elements like Strontium and Phosphorous show patterns very comparable to that of U$^{238}$. In contrast, the Mn distribution does not compare well to $^{238}$U, nor to $^{232}$Th.

Please note that single scattered datapoints towards higher values, best visible in the $^{88}$Sr and $^{232}$Th traces in Fig. 4 are considered to be analytical artefacts (outliers) occurring during laser ablation.






**Fig. 4: Stalagmite sample 2014#618 shows clear discoloration towards the right side of the stalagmite that was exposed to the sediment in which it was buried. Laser ablation trace element analysis along the 42 mm long transect shown on the picture demonstrates that Manganese is associated with the blackish layers that occur in the exterior portions of the stalagmite. It is further**

**clear that significant enrichment of U and Th occurs at the surface of the stalagmite that has been in contact with the sediment in which it was covered. Uranium declines gradually from the outer surface towards the center of the stalagmite. Sr shows a similar pattern to U. The Th content of the entire trace can be considered very low, with the exception of the increased Th concentration on the outer surface.**

**3.2    Observations per stalagmite level**

Visual and petrographic inspection showed that stalagmites growing from the top of unit H (specimens 2010#1, 2010#2 and 2012#5) were overall well preserved, presumably because they represent the only material that has not been covered by siliciclastic sediments. Some dissolution features are visible, but the typical encrustations and calcite discoloration, visible in

speleothem material from the older Scladina units, are absent in unit-H speleothem material.



A small stalagmite was recovered from unit 1B-BRUN. This unit has several smaller speleothem occurrences, which are mostly thin and not so well preserved in the siliciclastic sediments that cover them. Stalagmite 2014#206, however, appears comparatively well-preserved at visual inspection. 4 samples were taken for U-Th dating in the growth axis of this stalagmite. Stalagmite material from unit 4A shows signs of alteration, as indicated by common discolorations in stalagmite specimen

2010#14 for example, very similar to the diagenetically altered zones of the (older) stalagmite 2014#618 from lithostratigraphic Unit 6B. Looking in more detail, specimen 2010#14 has localized transparent intervals in the growth axis that lack discoloration and appear well-preserved. We selected these intervals specifically for sampling for U-Th dating. A second stalagmite specimen consisting of two pieces (2012#1&2) from lithostratigraphic unit 4A appears generally better preserved than specimen 2010#14. This stalagmite shows an alternation of milky and transparent calcite laminae that is commonly

observed in well-preserved speleothems, and in microscope view a columnar fabric (Fig. 5) that testifies of unaltered speleothem calcite (Frisia et al., 2000).

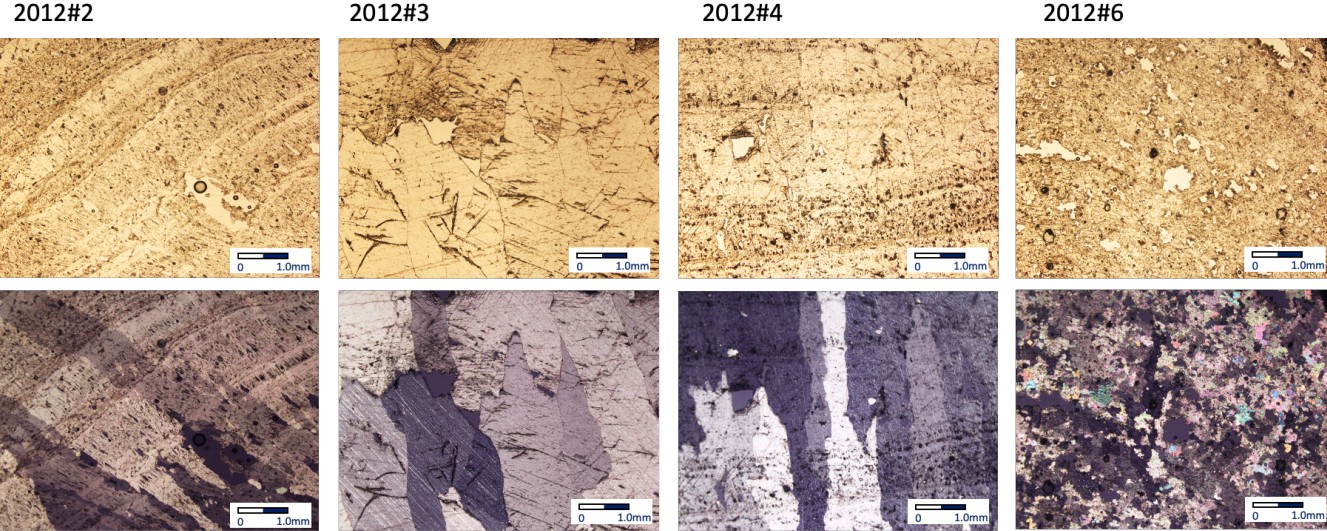

**Fig. 5: Some representative thin section images of the speleothem material studied. The top row are plain light (PL) images, with the growth direction generally upward. The bottom row are the same pictures, in cross polarized light (CPL). Specimens 2012#2 and 2012#4 show micro-scale growth banding of speleothem calcite in PL and cross-cutting columnar crystallographic fabric visible in CPL, which we interpret to reflect well-preserved speleothem calcite. Fluid inclusions in these images are shown as small black dots that generally line up along the growth bands. The images of specimen 2012#3 show no growth banding, relatively coarse crystals that are elongated in the growth direction, and a conspicuous scarcity of fluid inclusions. The images of specimen 2012#6 show a lack of growth banding and very small equidimensional crystals which we interpret to indicate diagenetic alteration of speleothem calcite. That specimen was discarded, and has not further been analysed in this study.**





Specimen 2010#6 is from the same stratigraphic interval as 2014#618 (lithostratigraphic unit 6B) and shows similar diagenetic features (yellow and whitish discoloration around the edges, and some conspicuous black thin laminae). We selectively sampled both specimens in a visually well-preserved zone in the central growth axis with laminae of alternating milky and transparent calcite where the typical diagenetic discoloration is not apparent.

Specimen 2012#4 is a flowstone from the lower half of unit XVII (Sous-Saint-Paul Cave) that consists largely of transparent

calcite with comparatively large crystal size. There is significant porosity in the horizontally layered portion of this flowstone, that may be of diagenetic origin. A small stalagmitic structure in the central part of this specimen is not porous. This stalagmitic structure shows a crystallography (Fig. 5) that we interpret to be unaltered, hence samples for U-Th dating and fluid inclusion isotope analysis were taken from that stalagmitic structure.

Specimen 2012#3 (at the base of unit XIX from Sous-Saint-Paul Cave) represents the stratigraphically oldest level in our

collection. It is a fragment of flowstone completely consisting of rather coarsely crystalline transparent calcite. Microscopic evaluation shows large calcite crystals which, apart from their unusual size perhaps, suggest no obvious diagenetic alteration (Fig. 5). Specimen 2012#3 further shows a conspicuous lack of fluid inclusions in the calcite crystals. The specimen was sampled along the growth axis in the center of the fragment collected.

**3.3    U-Th ages**

The results of U-Th dating are presented in Table 2 and Fig. 6. The speleothems from unit H, all yield Holocene ages, which generally confirms previously published U-Th ages (Gewelt et al., 1992). The stratigraphically oldest sample from the base of stalagmite 2012#5 from that speleothem complex, reaches 8.9 ka BP, and the composite record of stalagmites on top of unit

H seem to suggest near-continuous speleothem growth until the present day.

Stalagmite 2014#206 from layer 1B-BK (unit 1B-BRUN) produced four ages (Table 2). Two of which, at the base of the stalagmite, indicate growth in MIS-5a and another two higher up in the stalagmite correspond to MIS-3 (Fig. 6). Comparatively little stalagmite growth thickness between these two age clusters may suggest reduced or no growth in MIS-4.






Table 2

| Sample | DFT [mm] | Lab | $^{232}$Th [ng/g] | Uncertainty | $^{238}$U [µg/g] | Uncertainty | $(^{234}\mathrm{U}/^{238}\mathrm{U})$ | Uncertainty | $(^{230}\mathrm{Th}/^{238}\mathrm{U})$ | Uncertainty | $(^{230}\mathrm{Th}/^{232}\mathrm{Th})$ | Uncertainty | Uncorrected Age [ka BP] | Uncertainty | Corrected Age [ka BP] | Uncertainty [ka] |
|---|---|---|---|---|---|---|---|---|---|---|---|---|---|---|---|---|
| 2010#1 | 22 | Melbourne | 0.427 | 0.038 | 0.173 | 0.013 | 1.0417 | 0.0036 | 0.007765 | 0.00055 | 9.72 | 0.82 | 0.740 | 0.057 | 0.673 | 0.066 |
| 2010#1 | 102 | Melbourne | 0.238 | 0.023 | 0.209 | 0.016 | 1.0414 | 0.0029 | 0.00986 | 0.00060 | 27.2 | 2.2 | 0.974 | 0.063 | 0.943 | 0.065 |
| 2010#1 | 152 | Melbourne | 0.332 | 0.032 | 0.211 | 0.016 | 1.0414 | 0.0037 | 0.01127 | 0.00091 | 27.2 | 2.3 | 1.122 | 0.097 | 1.080 | 0.097 |
| 2010#1 | 256 | Melbourne | 0.240 | 0.025 | 0.190 | 0.014 | 1.0516 | 0.0030 | 0.01521 | 0.00045 | 37.8 | 2.2 | 1.524 | 0.048 | 1.491 | 0.051 |
| 2010#1 | 330 | Melbourne | 2.23 | 0.18 | 0.220 | 0.017 | 1.0542 | 0.0031 | 0.02410 | 0.00073 | 7.47 | 0.27 | 2.458 | 0.078 | 2.19 | 0.15 |
| 2010#1 | 418 | Melbourne | 0.782 | 0.069 | 0.214 | 0.016 | 1.0512 | 0.0038 | 0.0293 | 0.0013 | 25.2 | 1.4 | 3.02 | 0.13 | 2.92 | 0.15 |
| 2010#2 | 80 | Melbourne | 2.18 | 0.19 | 0.211 | 0.016 | 1.0422 | 0.0032 | 0.01310 | 0.00065 | 3.99 | 0.24 | 1.316 | 0.070 | 1.04 | 0.15 |
| 2010#2 | 170 | Melbourne | 0.872 | 0.076 | 0.212 | 0.016 | 1.0480 | 0.0037 | 0.01536 | 0.00088 | 11.71 | 0.81 | 1.546 | 0.092 | 1.44 | 0.11 |
| 2010#2 | 415 | Melbourne | 2.15 | 0.18 | 0.223 | 0.017 | 1.0476 | 0.0037 | 0.02999 | 0.00096 | 9.74 | 0.40 | 3.10 | 0.10 | 2.85 | 0.16 |
| 2010#2 | 530 | Melbourne | 17.3 | 1.5 | 0.187 | 0.014 | 1.0513 | 0.0090 | 0.0669 | 0.0044 | 2.27 | 0.17 | 7.11 | 0.48 | 4.6 | 1.3 |
| 2012#5 | 740 | Mainz JGU | 1.2056 | 0.0076 | 0.1911 | 0.0011 | 1.0358 | 0.0005 | 0.08226 | 0.00037 | 39.86 | 0.19 | 8.964 | 0.042 | 8.789 | 0.096 |
| 2012#5 | 740 | Melbourne | 40.7 | 3.2 | 0.203 | 0.015 | 1.0172 | 0.0035 | 0.1202 | 0.0017 | 1.878 | 0.033 | 13.64 | 0.22 | 8.0 | 2.9 |
| 2012#5 | 376 | Melbourne | 1.88 | 0.15 | 0.187 | 0.014 | 1.0400 | 0.0034 | 0.06576 | 0.00059 | 20.53 | 0.29 | 7.053 | 0.068 | 6.78 | 0.15 |
| 2012#5 | 0 | Melbourne | 1.26 | 0.10 | 0.196 | 0.015 | 1.0495 | 0.0039 | 0.03646 | 0.00049 | 17.84 | 0.32 | 3.788 | 0.055 | 3.62 | 0.10 |
| 2010#14 | 25 | Melbourne | 8.09 | 0.65 | 0.237 | 0.018 | 1.0769 | 0.0041 | 0.7199 | 0.0088 | 66.2 | 1.1 | 118.0 | 2.7 | 117.2 | 2.7 |
| 2010#14 | 97 | Melbourne | 0.833 | 0.075 | 0.254 | 0.019 | 1.1059 | 0.0066 | 0.7351 | 0.017 | 704 | 17 | 116.1 | 2.0 | 116.0 | 2.1 |
| 2010#14 | 150 | Melbourne | 3.15 | 0.26 | 0.326 | 0.025 | 1.1301 | 0.0047 | 0.7586 | 0.0077 | 246.5 | 4.5 | 117.5 | 2.2 | 117.3 | 2.3 |
| 2014#206 | 85 | XiAn | 8.91 | 0.18 | 0.23302 | 0.00037 | 1.0737 | 0.0017 | 0.5619 | 0.0012 | 44.93 | 0.90 | 79.86 | 0.33 | 78.86 | 0.57 |
| 2014#206 | 75 | XiAn | 6.67 | 0.13 | 0.29772 | 0.00057 | 1.0895 | 0.0018 | 0.5415 | 0.0014 | 73.8 | 1.5 | 74.01 | 0.32 | 73.42 | 0.42 |
| 2014#206 | 40 | XiAn | 11.91 | 0.24 | 0.21763 | 0.00031 | 1.3464 | 0.0019 | 0.5429 | 0.0013 | 30.33 | 0.61 | 54.84 | 0.20 | 53.72 | 0.50 |
| 2012#2 | 255 | Melbourne | 36.61 | 0.74 | 0.25621 | 0.00074 | 1.1042 | 0.0018 | 0.4608 | 0.0018 | 9.86 | 0.20 | 58.25 | 0.35 | 54.5 | 1.8 |
| 2012#2 | 310 | Melbourne | 0.298 | 0.025 | 0.191 | 0.014 | 1.0741 | 0.0030 | 0.7117 | 0.0030 | 1433 | 17 | 116.2 | 1.1 | 116.2 | 1.1 |
| 2012#2 | 410 | Melbourne | 0.375 | 0.030 | 0.208 | 0.016 | 1.0589 | 0.0021 | 0.7079 | 0.0029 | 1228 | 10 | 118.50 | 0.98 | 118.5 | 1.0 |
| 2012#1 | 244 | Mainz MPIC | 0.590 | 0.050 | 0.186 | 0.014 | 1.0609 | 0.0028 | 0.7165 | 0.0047 | 707 | 11 | 120.6 | 1.6 | 120.6 | 1.6 |
| 2012#1 | 172 | Mainz MPIC | 7.831 | 0.078 | 0.1895 | 0.0012 | 1.0768 | 0.0017 | 0.7279 | 0.0017 | 53.85 | 0.49 | 120.6 | 1.2 | 120.1 | 1.2 |
| 2012#1 | 84 | Mainz MPIC | 3.308 | 0.033 | 0.1810 | 0.0012 | 1.0656 | 0.0020 | 0.7124 | 0.0020 | 120.4 | 1.1 | 120.0 | 1.1 | 119.8 | 1.3 |
| 2012#1 | 2 | Mainz MPIC | 2.282 | 0.024 | 0.2160 | 0.0016 | 1.0709 | 0.0016 | 0.7222 | 0.0036 | 225.6 | 2.1 | 120.0 | 1.1 | 120.8 | 1.2 |
| 2010#6 | 70 | Mainz JGU | 11.66 | 0.12 | 0.2332 | 0.0016 | 1.0604 | 0.0014 | 0.7215 | 0.0044 | 40.84 | 0.42 | 122.3 | 1.4 | 120.8 | 1.6 |
| 2010#6 | 48 | Melbourne | 4.237 | 0.030 | 0.256 | 0.019 | 1.0736 | 0.0054 | 0.8624 | 0.0083 | 61.02 | 0.96 | 171.2 | 4.6 | 170.1 | 4.6 |
| 2010#6 | 35 | Mainz JGU | 11.35 | 0.92 | 0.2550 | 0.0016 | 1.07447 | 0.00033 | 0.8694 | 0.0033 | 634.3 | 3.1 | 174.0 | 1.6 | 173.9 | 1.6 |
| 2014#618-3 | 80 | Mainz MPIC | 1.2052 | 0.0085 | 0.2877 | 0.0018 | 1.09156 | 0.00042 | 0.8087 | 0.0029 | 142.7 | 1.0 | 142.7 | 1.0 | 142.2 | 1.1 |
| 2014#618-2-SV11 | | XiAn | 3.289 | 0.066 | 0.23568 | 0.00025 | 1.0549 | 0.0013 | 0.8934 | 0.0013 | 87.3 | 1.8 | 197.6 | 1.1 | 196.7 | 1.2 |
| 2014#618-2-SV12 | | XiAn | 7.37 | 0.15 | 0.2357 | 0.00029 | 1.0549 | 0.0013 | 0.8919 | 0.0013 | 1610 | 22 | 196.9 | 3.9 | 196.8 | 3.9 |
| 2012#3 SV-9 | 60 | XiAn | 3.9955 | 0.0055 | 0.2877 | 0.0018 | 1.07447 | 0.00033 | 0.8694 | 0.0033 | 7060 | 140 | 196.5 | 1.1 | 196.1 | 1.1 |
| 2012#3 SV-10 | 10 | XiAn | 0.1411 | 0.0029 | 0.29206 | 0.00032 | 1.0876 | 0.0012 | 1.1150 | 0.0017 | 7060 | 140 | 536 | +29 / -23 | 536 | +30 / -23 |
| 2012#3 | 50 | Mainz MPIC | 0.603 | 0.012 | 0.33958 | 0.00039 | 1.0822 | 0.0012 | 1.1001 | 0.0016 | 1894 | 38 | 475 | +16 / -14 | 475 | +16 / -14 |
| 2012#3 | 50 | Melbourne | 0.0722 | 0.0016 | 0.5491 | 0.0036 | 1.9982 | 0.011 | 1.092 | 0.011 | 25370 | 580 | 431 | +21 / -17 | 430 | +22 / -18 |
| 2012#3 | 35 | Melbourne | 0.141 | 0.012 | 0.381 | 0.029 | 1.0907 | 0.0027 | 1.031 | 0.0035 | 9379 | 100 | 430 | +29 / -24 | 363 | +31 / -23 |
| 2012#3 | 15 | Mainz MPIC | 0.213 | 0.018 | 0.240 | 0.018 | 1.0881 | 0.0034 | 1.0998 | 0.0049 | 3895 | 59 | 431 | +35 / -27 | 431 | +34 / -26 |
| 2012#3 | 15 | Melbourne | 0.0852 | 0.0018 | 0.2941 | 0.0020 | 1.0838 | 0.0023 | 1.1026 | 0.0100 | 11630 | 260 | 477 | +101 / -55 | 477 | +105 / -55 |
| 2012#4 | 84 | Melbourne | 0.157 | 0.013 | 0.276 | 0.021 | 1.0849 | 0.0024 | 1.0949 | 0.0057 | 6040 | 83 | 429 | +32 / -25 | 429 | +33 / -26 |
| 2012#4 | 48 | Melbourne | 9.49 | 0.81 | 0.192 | 0.014 | 1.1074 | 0.0029 | 1.0709 | 0.0052 | 67.8 | 1.0 | 303.6 | +9.3 / -8.6 | 302.5 | +9.4 / -8.6 |
| 2012#4 | 48 | Mainz MPIC | 5.73 | 0.47 | 0.147 | 0.011 | 1.1043 | 0.0023 | 1.0728 | 0.0052 | 86.50 | 0.95 | 312.0 | 9.2 | 311.1 | +9.4 / -8.4 |
| 2012#4 | 20 | Melbourne | 1.156 | 0.012 | 0.1475 | 0.010 | 1.1049 | 0.0052 | 1.0679 | 0.0059 | 416.6 | 4.1 | 300.4 | +9.4 / -8.6 | 300.2 | 8.8 |
| 2012#4 | 20 | Mainz MPIC | 0.703 | 0.059 | 0.184 | 0.014 | 1.1094 | 0.0026 | 1.0734 | 0.0046 | 883 | 11 | 303.7 | +8.2 / -7.7 | 303.7 | +8.4 / -7.6 |

**Table 2: Results of the $^{230}$Th/U-dating of the Scladina samples**







**Fig. 6:** The U-Th ages of the individual speleothems that were obtained in this study are plotted as coloured dots with their 2σ-uncertainty limits, and are projected onto global and regional climate proxy records. The global benthic $\delta^{18}$O probabilistic stack (Ahn et al., 2017) with the mean and 95% confidence interval given as a black solid line and grey areas, respectively. This global benthic $\delta^{18}$O stack provides the glacial-interglacial variability, whereby the glacial and interglacial (green bars) Marine Isotope Stages are defined by Lisiecki and Raymo (2005). The orange line indicates the Uk'37 sea surface temperature (SST) record from marine sediment core MD01-2443 off the Iberian Margin (Martrat et al., 2007). The high-resolution $\delta^{18}$O record of the North Greenland Ice Core Project (NGRIP) site is plotted as a blue line together with the Greenland stadials (GS) and interstadials (GI: grey bars) (Rasmussen et al., 2014; Seierstad et al., 2014).






Unit 4A stalagmites yield consistent ages, that all fit in MIS-5e, the warmest stage of MIS-5 (Table 2; Fig. 6). The oldest ages we captured for unit 4A are 121ka BP. The youngest age for the two unit-4A stalagmites analysed is 116 ka BP.

Two stalagmites from flowstone levels in stratigraphic unit 6B yield ages at 197ka BP for specimen 2014#618, and ages from 142 to 174ka BP for specimen 2010#6. The older ages can be attributed to the warm period MIS-7a, which fits the general

pattern that, in Scladina Cave, speleothems grow predominantly during interglacials. Interestingly, the younger age range of specimen 2010#6 places it in MIS-6, which is a glacial period.

The next speleothem level further down in the Scladina Cave stratigraphy, is in fact situated in the underlying Sous-Saint-Paul Cave, in unit XVII. For the sub-samples taken from flowstone sample 2012#4 we have consistent ages of 300-311ka BP at ±9ka 2σ uncertainties, suggesting the flowstone formed under interglacial conditions in MIS-9 (Fig. 6; Table 2).

The lowest speleothem level encountered in the Sous-Saint-Paul Cave (2012#3) again is a flowstone layer related to unit XIX. For this layer the ages plot between 363 and 536ka BP (Table 2), measured in three labs, roughly spanning the time period between MIS-14 and MIS-10. The ages are partly inversed, which is perhaps not surprising in view of the high and variable dating uncertainties (from 105 to 14ka at 2σ level) in a time interval that is so close to the limit of what can still be dated with the U-Th technique.


### 3.4 Stable isotope data of speleothem CaCO₃

Stable carbon and oxygen isotope data of speleothem calcite ($\delta^{13}C_{cc}$ and $\delta^{18}O_{cc}$ values) have been analysed, and are presented as a box-and-whisker plot in Fig. 7. What we see there is a distinct trend in stable isotope values through time with highest

values for $\delta^{13}C_{cc}$ and $\delta^{18}O_{cc}$ around MIS7 and 5. It is beyond the scope of the present paper to assess the detailed temporal stable isotope variation within the speleothem specimens studied, but in general, the averages of the specimens studied are within the range of values observed in the well-preserved Holocene speleothems for $\delta^{13}C_{cc}$ and $\delta^{18}O_{cc}$ which suggests that the interpretation of this range of values does not directly indicate a clear diagenetic overprinting of the original signal. Also the clear positive correlation between $\delta^{13}C_{cc}$ and $\delta^{18}O_{cc}$ data is commonly seen in well preserved speleothem material, and may

reflect changes in equilibrium conditions, indirectly related to water availability (Verheyden et al., 2008), or the coupled effect of temperature, rainfall isotope composition and rainfall amount on the one hand, and soil composition and activity on the other (Lachniet, 2009).



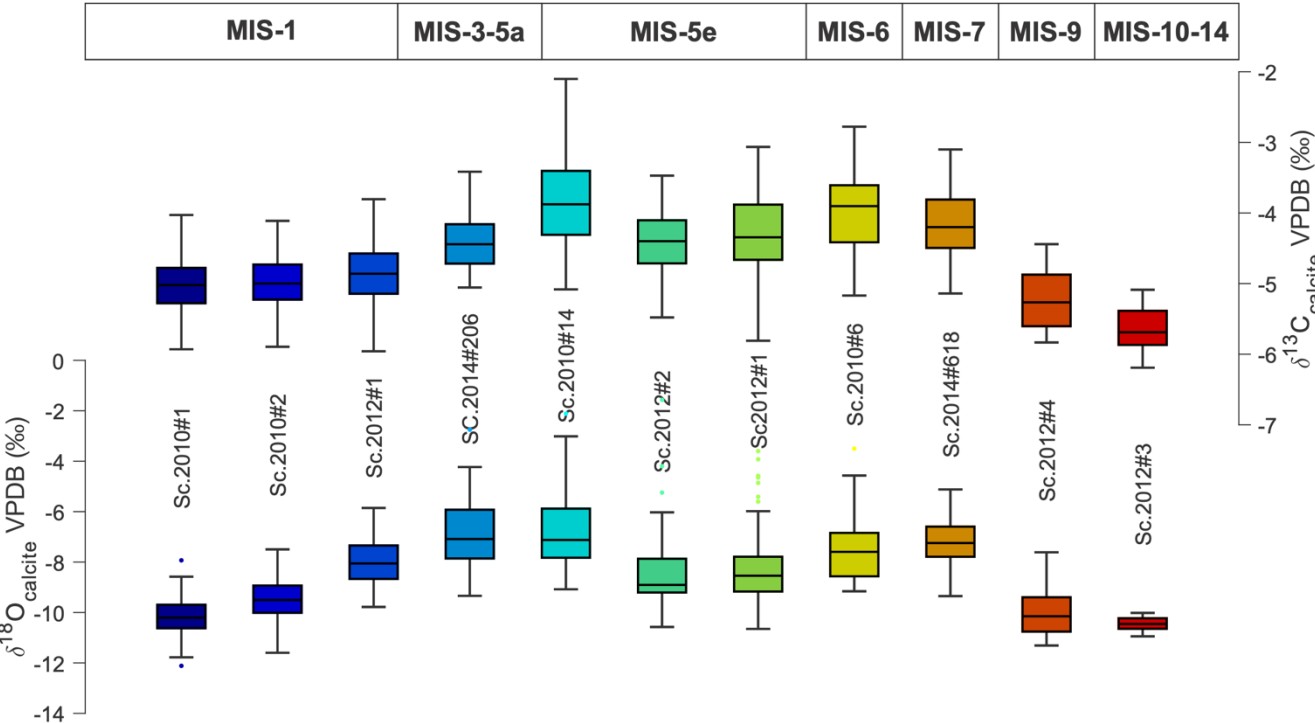

**Fig. 7: Box plots of stable isotope data of Scladina speleothem calcite show significant isotopic overlap between time intervals, but also clear parallel trends in average carbon and oxygen isotope values through time.**

### 3.5 Fluid inclusion stable isotope data

For all specimens studied we performed isotope analysis of fluid inclusion water on multiple subsamples. The isotope composition of fluid inclusion water in speleothem calcite ($\delta^2H_{fi}$ and $\delta^{18}O_{fi}$) is considered to represent that of the drip water in the cave at the time of speleothem growth, which in turn is a direct proxy for the isotope composition of (paleo)rainfall recharging the cave aquifer (Dennis et al., 2001; Van Breukelen et al., 2008). The oldest specimen, 2012#3, contained too little fluid inclusion water for analysis. Fluid inclusion isotope values from most other speleothems, presented in Table 3 and Fig. 8, fall on a trend with the majority of the samples plotting around an endmember close to the Global Meteoric Water Line (GMWL).





| Sample | DFT | $\delta^{13}C_{cc}$ | $\delta^{18}O_{cc}$ | CaCO$_3$ weight | yield | rel yield | $\delta^{18}O_{fi}$ | $\delta^2H_{fi}$ | calculated T |
|---|---|---|---|---|---|---|---|---|---|
| | cm | VPDB | VPDB | g | µl | µl/g | VSMOW | VSMOW | ºC |
| 2012#5 | 29 | - | - | 0.50 | 0.27 | 0.54 | -4.97 | -43.08 | - |
| 2012#5 | 20 | -8.8 | -4.4 | 0.60 | 0.25 | 0.42 | -6.05 | -46.66 | 11.1 |
| 2012#5 | 2 | -6.7 | -3.9 | 0.40 | 0.26 | 0.65 | -5.94 | -44.69 | 8.8 |
| 2012#5 | 2 | -7.0 | -4.2 | 0.40 | 0.20 | 0.50 | -6.95 | -47.01 | 5.3 |
| 2012#5 | 2 | -6.9 | -4.2 | 0.30 | 0.18 | 0.60 | -6.04 | -45.49 | 9.7 |
| 2012#5 | 25 | -6.8 | -3.8 | 0.40 | 0.09 | 0.23 | -6.86 | -46.42 | 4.1 |
| 2012#5 | 10 | -7.8 | -4.5 | 0.40 | 0.18 | 0.45 | -4.54 | -42.76 | 19.4 |
| 2012#5 | 41 | -8.5 | -4.6 | 0.50 | 0.40 | 0.80 | -6.21 | -43.83 | 10.9 |
| 2012#5 | 10 | -7.8 | -4.6 | 0.40 | 0.21 | 0.53 | -4.88 | -44.44 | 17.9 |
| 2012#5 | 5 | -8.1 | -4.2 | 0.50 | 0.30 | 0.60 | -5.21 | -43.12 | 14.0 |
| 2012#5 | 29 | -8.5 | -4.6 | 0.60 | 0.48 | 0.80 | -5.21 | -43.89 | 16.0 |
| 2012#5 | 45 | -7.5 | -4.2 | 0.46 | 0.21 | 0.46 | -2.89 | -39.84 | 26.4 |
| 2012#5 | 50 | -8.6 | -4.8 | 0.70 | 0.28 | 0.40 | -6.41 | -44.29 | 11.3 |
| 2012#5 | 45.5 | -8.4 | -4.6 | 0.72 | 0.20 | 0.28 | -5.29 | -41.6 | 15.6 |
| 2012#5 | 60 | -7.8 | -4.8 | 0.38 | 0.33 | 0.87 | -7.04 | -46.86 | 7.9 |
| 2012#5 | 70 | -8.4 | -4.9 | 0.39 | 0.07 | 0.18 | -4.73 | -41.35 | 20.5 |
| 2012#5 | 60 | -7.6 | -4.6 | 0.40 | 0.41 | 1.03 | -6.96 | -46.51 | 7.4 |
| 2012#5 | 70 | -8.8 | -4.9 | 0.57 | 0.14 | 0.25 | -3.907 | -40.36 | 25.0 |
| 2010#2 | 50 | -8.0 | -4.4 | 0.68 | *0.05* | 0.07 | - | - | - |
| 2010#2 | 7 | -10.0 | -5.1 | 0.90 | 0.33 | 0.37 | -6.11 | -42.12 | 13.9 |
| 2010#1 | 37 | -10.3 | -5.5 | 0.70 | 0.34 | | -6.62 | -47.13 | 13.5 |
| 2010#2 | 24 | -10.0 | -5.3 | 0.81 | 0.29 | 0.36 | -6.96 | -44.68 | 10.6 |
| 2010#1 | 11 | -8.6 | -4.6 | 0.56 | *0.05* | 0.09 | - | - | - |
| 2010#1 | 27 | -10.9 | -5.2 | 0.76 | 0.28 | 0.37 | -7 | -44.38 | 10.1 |
| 2012#4 | 2 | -9.5 | -5.0 | 0.81 | 1.01 | 1.25 | -7.67 | -53.62 | 5.8 |
| 2012#4 | 8 | -8.7 | -4.9 | 0.46 | 1.02 | 2.22 | -7.56 | -53.8 | 5.8 |
| 2012#4 | 2 | -9.7 | -5.0 | 0.27 | 0.51 | 1.89 | -7.6 | -54.39 | 6.3 |
| 2012#4 | 8 | -9.0 | -4.9 | 0.33 | 0.54 | 1.64 | -7.98 | -55.49 | 3.9 |
| 2012#1 | 21 | -7.0 | -3.4 | 0.45 | *0.03* | 0.07 | - | - | - |
| 2012#1 | 2 | -9.4 | -4.6 | 0.87 | 0.29 | 0.33 | -6.89 | -46.43 | 7.8 |
| 2012#1 | 10 | -8.6 | -4.3 | 0.95 | 0.44 | 0.46 | -6.68 | -44 | 7.1 |
| 2012#1 | 21 | -7.6 | -3.6 | 0.98 | 0.20 | 0.20 | -4.74 | -41.09 | 13.3 |
| 2012#2 | 38 | -7.4 | -3.9 | 0.90 | 0.43 | 0.48 | -6.08 | -43.49 | 8.4 |
| 2012#2 | 19 | -5.2 | -3.2 | 0.89 | 0.31 | 0.35 | -5.12 | -41.9 | 9.3 |
| 2012#2 | 4 | -9.0 | -4.2 | 0.75 | 0.40 | 0.53 | -6.78 | -46.92 | 6.4 |
| 2012#2 | 31 | -8.1 | -4.2 | 0.78 | 0.41 | 0.53 | -6.53 | -44.01 | 7.4 |
| 2010#14 | 30 | -8.6 | -4.5 | 0.79 | 0.18 | 0.23 | -4.25 | -41.21 | 20.5 |
| 2010#14 | 19 | -8.4 | -4.7 | 0.88 | 0.15 | 0.17 | -5.91 | -44.44 | 13.1 |
| 2010#14 some visible diagenesis | 22 | -5.3 | -3.2 | 0.74 | *0.02* | 0.03 | - | - | - |
| 2010#6 | 10 | -8.8 | -4.7 | 0.88 | 0.41 | 0.47 | -6.17 | -44.07 | 11.7 |
| 2010#6 | 4 | -8.2 | -4.1 | 0.80 | 0.22 | 0.28 | -1.27 | -37.77 | 35.6 |
| 2010#6 | 13 | -7.9 | -3.9 | 0.84 | 0.17 | 0.20 | -0.65 | -36.58 | 37.5 |
| 2010#6 | 6 | -8.0 | -4.1 | 0.93 | 0.51 | 0.55 | -6.35 | -46.06 | 7.6 |
| 2010#6 | 4 | -8.1 | -4.1 | 0.80 | *0.05* | 0.06 | - | - | - |
| 2014#618-3 base | 10 | -7.9 | -4.1 | 0.80 | 0.44 | 0.55 | -6.25 | -44.52 | 8.3 |
| 2014#618-3 Middle | 5 | -7.0 | -4.3 | 0.79 | 0.56 | 0.71 | -6.47 | -44.61 | 8.1 |
| 2014#618-3 side some visible diagenesis | - | -5.41 | -3.62 | 0.68 | *0.02* | 0.03 | - | - | - |
| 2014#618-3 top side significant visible diagenesis | - | - | - | 1.10 | failed | - | - | - | - |
| 2014#206 base of stal | 90 | -7.01 | -4.45 | 0.50 | 0.93 | 1.86 | -6.75 | -49.29 | 7.6 |
| 2014#206 | 24 | -7.97 | -4.88 | 0.50 | 0.64 | 1.28 | -5.45 | -47.48 | 16.3 |
| | | | | | | | | | |
| drip water sample Scladina Cave | | | | | | | -6.48 | -42.40 | |

**Table 3: Stable isotope data of Scladina Cave fluid inclusion water samples, and one drip water sample (bottom). Also indicated are the stable isotope values of the host calcite of the fluid inclusion samples, and the calculated temperatures from the paired fluid inclusion and host calcite oxygen isotope ratios. This calculation is based on the (Tremaine et al., 2011) equation. DFT = distance from top.**





**Fig. 8: fluid inclusion isotope data of Scladina speleothems shown as round plot symbols. GNIP rainfall isotope data for Liege in the time period between 1966 and 1970 are indicated as grey crosses. The GMWL is shown as a blue line, the local (Liege) MWL as the dotted blue line. Parallel grey lines indicate the GMWL+5 to GMWL-5 envelope around the GMWL. Annually averaged rainwater isotope value and the drip water isotope composition in the cave is shown as diamonds.**

Part of the data trend away from the GMWL at a relatively low angle towards higher $\delta^2H_{fi}$ and $\delta^{18}O_{fi}$ values (Fig. 8). This trend seems to occur in data from all studied time intervals, with one clear exception. This exception is specimen 2012#4, dated at MIS 9 age, that displays the lowest $\delta^2H_{fi}$ and $\delta^{18}O_{fi}$ data of all material analysed and plots in a tightly constrained field close to the GMWL (Fig. 8).

The fluid inclusion isotope data that plot near the GMWL have isotope values close to the annually-averaged isotope composition of modern regional rainwater ($\delta^2H_p$ and $\delta^{18}O_p$), of -47.3 and -6.8 ‰ (VSMOW) for $\delta^2H$ and $\delta^{18}O$ respectively. Rainfall data is provided by the Global Network of Isotopes in Precipitation (IAEA/WMO, 2023) database for the Liege Weather station (for the years 1966-1970). A single drip water sample from Scladina Cave (taken in 2010) gave a $\delta^2H$ value





of -42.4 ‰ and a $\delta^{18}O$ value of -6.5 ‰, which is also close to the weighted-average rainfall isotope values for Liege, as documented in the Global Network of Isotopes in Precipitation database.

## 4    Discussion

### 4.1    Diagenetic screening of the samples


While there is no doubt that diagenesis has at least partly altered the Scladina Cave speleothems, we have made several observations indicating that diagenesis is not pervasive, and that in the samples we selected for U-Th analysis we successfully avoided diagenetically altered parts of the speleothems. Particularly the Laser Ablation ICP-MS trace element patterns (Fig. 4) suggest significant diagenetic alteration on the stalagmite outer surface, while the calcite in the growth axis shows trace

element concentrations that fit unaltered speleothem calcite. The different gradients of $^{238}U$ and $^{232}Th$ from the diagenetic surface towards the unaltered growth axis are interpreted to reflect the contrasting solution chemistry of U and Th. U can move in dissolved form with diagenetic fluids into the speleothem, while the less soluble Th, stays behind in particulate or colloidal state, on the outside. Elements like Strontium, also readily mobile in solution, show very comparable concentration trends to Uranium, suggesting that, once buried in detrital cave sediments, a dissolution-reprecipitation diagenetic front penetrated into

the speleothems from the surface inwards, altering trace element patterns. The elevated phosphorous content that line up well with the changes in U and Sr likely result from diagenetic mineralization of bat guano phosphate present in the detrital sediment matrix (Audra et al., 2021).

One more observation is that well before the transect reaches the growth axis of the speleothem, the trace element concentrations stabilize at values which are typical of well-preserved speleothem calcite. It appears therefore

that the speleothem calcite we selected for U-Th analysis is sufficiently well-preserved to yield good radiometric ages.

One more observation that follows from the trace element data is that the black laminae, occurring in some of the specimens studied, contain Manganese. These Manganese enrichments do not line up with the diagenetic patterns in Sr, U, and P. If these Mn enrichments are diagenetic, it appears that they are not formed by the same diagenetic process. In any case, we stayed away from these dark laminae when sampling for U-Th analysis.

This careful diagenetic screening of speleothem samples has led to what appears to be a relatively consistent U-Th chronology of the Scladina speleothem sequence. When properly applying $^{232}Th$ corrections and considering the analytical uncertainties that result from that, no age reversals are present in the dataset, with the exception of the samples from the oldest flowstone level 2012#3, which are very close to the dating limit of the U-Th technique. Where different specimens from a single stalagmite horizon were analysed, the resulting ages appear generally consistent, also between the different labs where the

dating took place.



## 4.2 Fluid inclusion isotope data

The presence of what appears to be well-preserved fluid inclusion water plotting isotopically close to the meteoric water line,
supports limited or absent diagenetic alteration of the selected speleothem material. In such "low water-rock ratio" systems any diagenetic alteration will impact the fluid inclusion isotope composition more readily than it will impact the calcite (Uemura et al., 2020). The observation that a significant part of the fluid inclusions trend away from the GMWL (Fig. 8), may suggest evaporation to have occurred in the cave (Warken et al., 2022). However, the slope of the evaporation line for Scladina appears too low to represent evaporation in a cave environment, but rather resembles diffusive water loss from speleothem
calcite as an analytical artefact, reported in a recent study by Fernandez et al. (2023). We postulate that, in the case of Scladina Cave, diagenesis of speleothem calcite weakened fluid inclusion stability, facilitating diffusion of water out of the sample before analysis.

To confirm this, we combined the oxygen isotope composition of the fluid inclusions ($\delta^{18}O_{fi}$) with that of the corresponding host calcite ($\delta^{18}O_c$), to calculate past cave temperatures based on the assumption of 1) isotope equilibrium during formation,
and 2) absence of post-depositional alteration of fluid inclusion water or calcite (Table 3). Based on a widely accepted calculation specifically made for speleothems (Tremaine et al., 2011), these data show that fluid inclusion isotope values plotting further away from the GMWL lead to unrealistically high temperatures (Fig. B1), which supports that these fluid inclusion isotope values indeed are no longer original. Most of the data plotting close to the GMWL, however, result in paleotemperatures in the range of the modern cave temperatures, suggesting, in good agreement with the observations by
(Fernandez et al., 2023), that these fluid inclusion and calcite isotope values are essentially unaltered, and provide the original rainwater isotope values.

Projecting the data back onto the GMWL along the diffusion slope, one can reconstruct the original rainwater isotope signatures throughout the Scladina speleothem stratigraphy (Fernandez et al., 2023). This exercise shows how isotope values in MIS7, MIS5 and the Holocene were quite comparable at approximately -47‰ and -7‰ for $\delta^2H_{fi}$ and $\delta^{18}O_{fi}$ respectively, which
compares well to present day rainfall and dripwater isotope values, suggesting remarkable similarity in interglacial hydrological conditions through the past 200ka. In contrast, reconstructed rainwater of MIS9 is on the GMWL, but are lower by ~8‰ in $\delta^2H_{fi}$ values and by ~1‰ in $\delta^{18}O_{fi}$ values.

## 4.3 Paleoclimatological backdrop of speleothem growth in Scladina Cave




Based on this robust age framework for the horizons of speleothem growth in Scladina Cave it is now possible to consider the paleoclimatological backdrop of speleothem growth at this site in more detail than previously possible. Lining-up the Scladina speleothem ages along the late Quaternary Probabilistic Benthic Stack (Ahn et al. 2017) shows that speleothem growth in Scladina Cave generally took place during Interglacial Periods. (Fig. 6). During Glacial Periods, sedimentation in Scladina

Cave was dominated by siliciclastic influx (Pirson et al., 2014b).

Our age constraints for the Holocene unit H speleothem complex indicate that growth started at ~9ka. This is the time of the temperature rise in NW Europe, coming out of the Younger Dryas cold interval, approaching the warmest temperatures of the Holocene (Kaufman and Mckay, 2022). The southern North Sea flooded, disconnecting the British Isles from the mainland and providing a nearby moisture source for NW Europe. At that time, NW Europe was reaching the optimum of deciduous

forest cover (Marquer et al., 2014) also reflected in a strong increase in arboreal pollen in the Eifel Maar sequence (Riechelmann et al., 2023). Therewith, climate in NW Europe  provided ideal conditions for speleothem growth. Further climate variation took place throughout the Holocene, but our observation of (near) continuous speleothem growth through the Holocene in Scladina Cave suggests that hydroclimate in this part of Belgium generally supported speleothem growth, as it still does today.

The Scladina Cave speleothem record of stratigraphic unit 4A, corresponding to MIS5e (~ Eemian Interglacial), starts at ~ 121ka, which is distinctly younger than the ~129ka start of MIS-5e in speleothem records from southern Europe (Drysdale et al., 2005), or the ~125ka observed in nearby records in the Belgian Ardenne (Vansteenberge et al., 2016). This starting age likely has little paleoclimatological significance, as it is coming from unit 4A-CHE, which constitutes a large gully eroding the in situ stalagmitic floor from underlying unit 4A-IP. The ages obtained from the unit-4A-IP stalagmite suggest the sampled

speleothem did not record the lowermost part of the Eemian. The scarcity of in-situ stalagmites in this interval is presumably due to the complex sedimentary dynamics following unit 4A speleothem growth, including strong erosion phases (Pirson, 2007, 2014; Pirson et al., 2014b). It is therefore not a surprise that we don't capture the onset of the growth phases in the stalagmites collected. When the tops of reworked speleothems are preserved, the timing of growth cessation may still have paleoclimatological relevance. For example, the youngest ages of Scladina MIS5e speleothems, at ~116ka, coincide with the

transition towards the considerably cooler MIS-5d period. This Late Eemian growth stop in the Scladina record may also coincide with the Late Eemian aridity Pulse as observed in German Maar lakes (Sirocko et al., 2005) and previously also identified in other Belgian speleothems (Vansteenberge et al., 2019).

Stalagmite 2012#618 from unit 6B and flowstone 2012#3 from unit XVII also date to the later stages of their matching interglacial periods (MIS7 and MIS9 respectively), suggesting that, here too, we may capture the climate deterioration at the

end of the warm periods in the cessation of active speleothem growth in our samples, although it must be noted that growth cessation can also be caused by changes in dripwater trajectories that are not controlled by climate (Lechleitner et al., 2018).

Ages for the oldest speleothem (flowstone) material in this record have higher age uncertainties, and can therewith not be tied to a single interglacial period. All we can say is that growth occurred in the time period from MIS-10 to MIS-14 based on the U-Th data we collected for these older samples.




Finally, two speleothems have ages that suggest deposition in what are generally considered not to be interglacial conditions. Stalagmite 2010#6 has U-Th ages ranging from 174ka to 142ka, which corresponds to Glacial period MIS-6. Growth of particularly the older part of this specimen coincided with wetter conditions in Southern Europe, and the well-documented Mediterranean Sapropel S6 event that relates to a maximum in Northern hemisphere summer insolation (Bard et al., 2002;

Kroon et al., 1998; Wainer et al., 2013). Stalagmite 2010#6 may suggest that these wet conditions occurred further North in Europe as well. Even though Western Europe and the Mediterranean region experienced relatively wet conditions at that time, global sea level was significantly below the modern values (Bard et al., 2002). This appears to indicate that speleothem growth in Scladina Cave did not necessarily require full interglacial settings in the past. Favourable hydrological conditions around 170-174ka could perhaps relate to comparatively high sea surface temperatures (SST's) at the Iberian margin at that point in

time (Martrat et al., 2007). In contrast, at 142ka, the youngest age of stalagmite 2010#6, Iberian margin SST's are low, coinciding with the maximum ice extent in Northern Europe (Ahn et al., 2017). Growth conditions for speleothems in Scladina Cave must have been rather unfavourable at 142ka BP, and we are also not aware of other speleothem occurrences of this age in Northern Europe. Since some signs of diagenetic overprint have been observed in stalagmite 2010#6, we therefore suspect this particular age may be inaccurate.

Stalagmite 2014#206 is another specimen that grew under non-interglacial conditions. It has ages of ~78-74ka BP at the base (corresponding to MIS-5a), and ages around 53 ka BP towards the top (corresponding to MIS-3).

This interval is characterized by millennial-scale Dansgaard Oeschger variability (Rasmussen et al., 2014), which is well documented in Greenland Ice cores, whereby Greenland interstadials (warmer periods) would have been more likely to have favoured speleothem growth conditions in Scladina Cave. The oldest age of 2014#206 correspond to the transition from long

interstadial GI-21 to Greenland Stadial GS-20; (Rasmussen et al., 2014), and the younger ages of 2014#206 correspond to pronounced interstadial GI-14. (Fig. 6). Regardless of their assignment to Greenland stadials or interstadials, all measured ages of speleothem 2014#206 correspond to relatively high SST's at the Iberian margin, that are to some extent decoupled from Dansgaard Oeschger variability recorded in the NGRIP ice core for this time interval (Martrat et al., 2007). Particularly the interval around 53 ka BP also corresponds to a clear peak in tree pollen, and other forest indicators in the Eifel Maar record

(Riechelmann et al., 2023). For both the interval around 76ka and the interval around 53ka, speleothem growth occurs at several other European sites (Lechleitner et al., 2018; Peral et al., 2024; Riechelmann et al., 2023).

## 4.4 Stable isotope evolution of speleothem calcite through the Scladina time series

The pattern of stable isotope variation through the depositional history of Scladina Cave shows a general trend of increasing $\delta^{13}C_{cc}$ and $\delta^{18}O_{cc}$ values from the oldest speleothem deposits towards MIS 5e, after which the trend reverses towards lower isotope ratios again in the Holocene (Fig. 7). This trend is quite similar in both oxygen and carbon isotopes and appears to be





a robust feature in the material analysed. While such a trend can have several causes, the parallel variation of $\delta^{13}$C and $\delta^{18}$O

values of speleothem calcite is commonly interpreted in terms of a drier climate for the higher isotope values, and wetter

climate for the lower isotope values (Lachniet, 2009; Mcdermott, 2004). This would then suggest that driest interglacial

conditions occurred around MIS-5e and MIS-7. To what extent changes in the seasonal balance of rainfall through time could

have contributed to these isotope trends cannot be resolved from our dataset, although the data may suggest that the

conspicuously low fluid inclusion isotope values for stalagmite 2012#4 (MIS-9) could have resulted from an increased

contribution of winter rainfall (which nowadays has the lower isotope values).


### 4.5   New U-Th ages in comparison to the existing Scladina Cave age model

The new ages we present here compare favourably to the existing age model of Scladina Cave (Pirson et al., 2014b).

Methodological progress made over the past decades clearly comes forward in the direct comparison to previously published

U-Th ages (Gewelt et al., 1992) that are generally in the same range as our ages, but with much larger uncertainties. Our data

allow for several changes and additions to the existing age model.

First, the ages we provide for the lower part of the Scladina stratigraphic sequence chart new territory. The oldest units in the

Scladina Cave sequence that we could date are from a speleothem complex in unit 6B. At an age of 197ka BP, these

speleothems correlate to the latest part of MIS-7, while it has previously been systematically positioned inside MIS 5 (Bastin,

1992; Bonjean, 1998; Cordy and Bastin, 1992; Pirson et al., 2014b). These new ages demonstrate that the older part of the

Scladina sequence belongs to the Middle Pleistocene, and more specifically to the Late Saalian.

Furthermore, the ages of the unit 4A-IP stalagmites unequivocally indicate that they grew in MIS-5e, equivalent to the Eemian

period (Shackleton et al., 2003), while in the previous age model, these were attributed to MIS-5a or MIS-5c (Pirson et al.,

2014b). This has implications for the currently accepted age of the Scladina 1-4A Juvenile Neandertal remains, as these were

found in a stratigraphic unit directly overlying this stalagmitic floor. Even though the reworked nature of the Neandertal fossils

(Pirson et al., 2014a) as well as the lack of precision in the chronology of the sequence situated immediately above unit 4A-IP

(Pirson, 2014) prevent firm conclusions, the new ages make it more logical that the deposition of the Neandertal remains

predate MIS-5a, and more probably belong to MIS 5d.

These ages further suggest that the Middle Palaeolithic assemblage from unit 5 (Bonjean et al., 2014), that so far was dated at

130 ± 20 ka (Huxtable and Aitken, 1992) and positioned inside MIS 5 (Pirson et al., 2014b), predates MIS-5e and postdates

the brief climate amelioration dated at 174 ka BP in MIS-6 (recorded in stalagmite 2010#6 from stratigraphical unit 6B). In

this interpretation, we reject the 142ka BP age in the top section of speleothem 2010#6, on the suspicion that this age is

diagenetically compromised. The stratigraphical placement of the unit 5 archaeological assemblage in MIS 6 is also in better

agreement with the Infrared-stimulated Luminescence (IRSL) derived age of 153 ± 15 ka obtained from unit 4B, overlying

unit 5 (Pirson et al., 2014b).

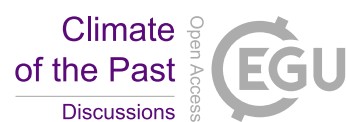

Finally, our results allow for the first time to assign ages to the stratigraphy of the underlying Sous-Saint-Paul Cave, showing that this sequence developed during Middle Pleistocene. Adding the sedimentary record of the underlying Sous-Saint-Paul Cave to that of Scladina Cave extends the complete sedimentary sequence down to at least MIS-10 or perhaps even to MIS-

14, which makes this one of the stratigraphically longer cave sedimentary records in North-West Europe.

## 5    Conclusions

Based on careful sample screening we identified speleothems with little to no diagenetic alteration from Sous-Saint-Paul and Scladina Caves, for U-Th dating. U-Th ages of these well-preserved speleothems confirm that their formation in these two

connected caves predominantly occurred in warm (interglacial or interstadial) periods of the late Quaternary. Intercalated siliciclastic sediments were the dominant mode of deposition in colder periods. The data provide a robust age framework for Scladina Cave, and add precisely dated stratigraphic anchor points to the younger part of the sequence, that improve the previous age model. One notable consequence of the new U-Th ages is that a juvenile Neandertal mandible, previously placed in Marine Isotope Stage 5a or 5b, could now potentially date back to Marine Isotope Stage 5d. Furthermore, new U-Th

analyses, provide Middle Pleistocene ages for the previously undated lower part of the Scladina sequence, and the underlying Sous-Saint-Paul sequence. All ages combined suggest this system to be one of the longer fossil-rich cave sedimentary sequences in NW Europe.

Fluid inclusion isotope data of the speleothem samples predominantly plot close to the GMWL, and are in good agreement with the isotope composition of modern rainfall and dripwater in the cave. This suggests that hydroclimatic conditions

conductive to speleothem growth in past warm periods were not very different from today. From a different perspective, these fluid inclusion isotope data also provide a clear indication of the good preservation of most of the speleothem material selected, as fluid inclusions are considered to be very sensitive to diagenetic alteration. A minority of the fluid inclusion data trends away from the GMWL, suggesting that these data were from potentially less well-preserved samples. This serves to remind us of the importance of careful sample selection in cave settings where burial in siliciclastic sediments has led to diagenetic

overprints on speleothem calcite.





## Appendices:

## Appendix A

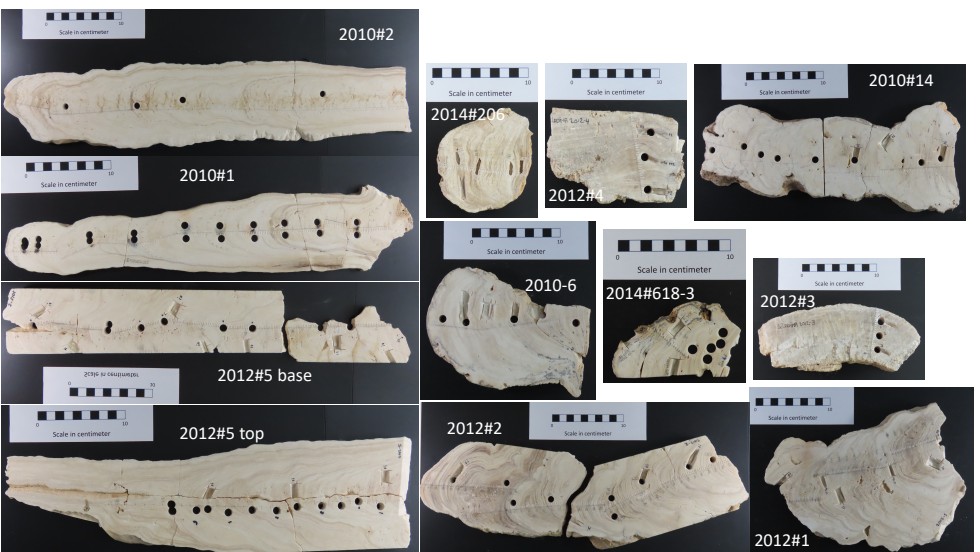

**Fig. A1: Overview of slabs cut from the stalagmite specimens studied**

## Appendix B

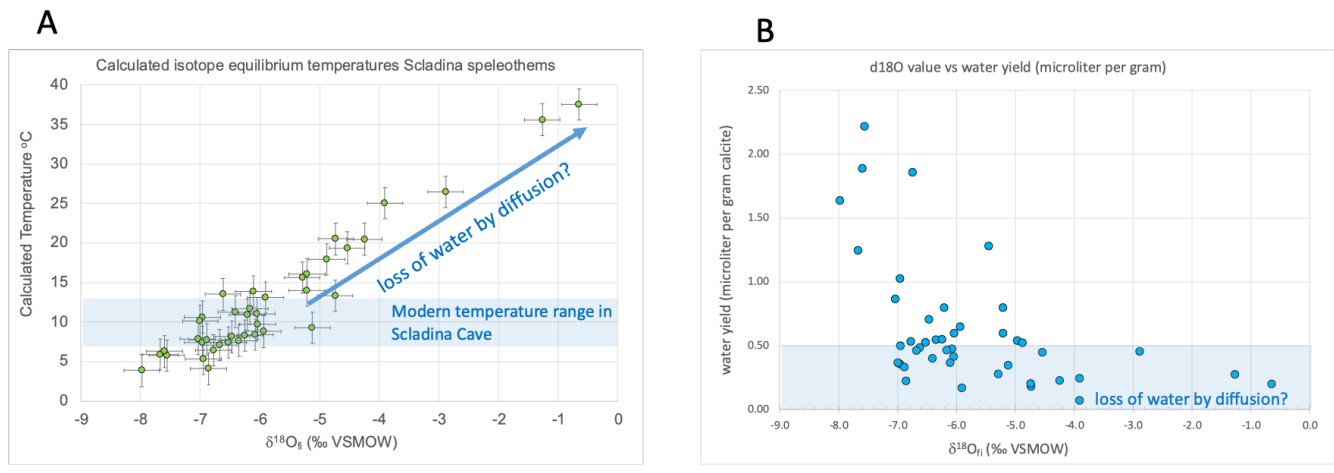

**Fig. B1 A) Isotope equilibrium temperatures calculated from paired oxygen isotope values of fluid inclusions and their host calcite. The blue shaded box spans the modern seasonal temperature variation measured in the cave. Higher $d^{18}O_{fi}$ values lead to unrealistically high temperatures, suggesting that the process that leads the fluid inclusion values away from the GMWL sends the fluid inclusions and their host calcite out of isotopic equilibrium. B) Cross plot showing that lower water yield samples tend to have higher $d^{18}O_{fi}$ values. Blue zone indicates samples with lower than 0.5 microliter per gram extraction yields which we postulate to be the samples that may have lost water through diffusion, leading the remaining water to be isotopically fractionated towards higher $d^{18}O_{fi}$ values.**



**Data availability:**


All data for this paper that are not listed in the tables, will be made available in the Edmond database:

https://edmond.mpg.de/dataverse/edmond

**Author contribution**


H.B.V coordinated the project in which A.J.v.N., J.R. M.v.d.D, and D.B. performed lab work and analyses. Samples were collected and placed in the stratigraphical context by D.B., S.P. K.d.M and G.A. J.H., D.S., M.W. S.V., H.C and X.J. provided U-Th analyses. J.v.d.L., D.B. and S.P. provided material for figures. H.B.V. prepared the manuscript with contributions from all co-authors.


**Competing interests**

The authors declare that they have no conflict of interest

**Acknowledgements**

This study is the cumulative result of many years of research effort of students at Vrije Universiteit Amsterdam, and later at the Max Planck Institut für Chemie, and the Johannes Gutenberg Universität, Mainz. This project would not even have started if it weren't for the enthusiastic support by Dick Kroon, at that point in time Professor at the Earth Sciences Department at

Vrije Universiteit Amsterdam. With his broad interest in new paleoceanographical and paleoclimatological proxy systems and techniques Dick supported and motivated HV to initiate a line of speleothem research that has been developing until today. Special acknowledgement goes to Laura García Soler, one of the students who contributed to this study. We have unfortunately not been able to reach her to offer co-authorship on this ms.

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
