# Peer review of "Improving the age constraints on the archeological record in Scladina Cave (Belgium): new speleothem U-Th ages."

_Climate of the Past, 2024_

## Author Response (AR1)

Reviewer 1:

This manuscript is focused on a new dating exercise of an existing archaeological sequence in Scladina Cave, Belgium. The authors provide new U-Th dates that result more accurate than previous ones, particularly due to an exhaustive examination of speleothem samples using trace elements and fluid inclusions to select the best intervals. Although my review is positive and consider this manuscript an interesting piece of work and an advance in the knowledge of an important and classic sequence, I have some comments and requests that should be considered before publication.

1.   I miss more discussion on how the U-Th technique has improved in the last decades and how these improvements have a consequence in a better chronology of the considered sequence. This is not explained and would be useful. Besides, the quality of the U-Th dates has to be evaluated according to Table 2 numbers and not once checking if they coincide or not with other studies (ej. line 484 where a date is discarded because there are not other examples of speleothem grow in Europe).

*We havesupplied information on the improvements of the U-Th technique. The main difference is that in the 1980's and early 1990's, when the earlier U-Th analyses were performed, the newest analytical technique available was Thermal Ionistation Mass Spectrometry, which was not as precise as the currently available "Multicollector Inductively Coupled Plasma Mass Spectrometry" (MC-ICP-MS) technique. The latter instrumentation is what is used in the labs contributing to this study.*

2.   I also miss more information about which diagenetic processes have been detected or approached by trace elements and isotopes. It is not clear to me what type of alterations (recrystallization? Fluid movements?) are to be avoided for reaching a better dating and why. In fact, some of the explanations are about discoloration or black layers outside the growh axis where sampling is carried out. Also it is said that samples were covered by siliciclastic sediments (line 244), how does it affect the speleothems? Please, provide references of similiar studies.

*We understand that more clarity is needed here. The diagenetic processes in carbonate rocks are complex, and in speleothems not so widely studied. The processes most studied in speleothems are the conversion from aragonite to calcite (for originally aragonitic speleothems, which is not the case here) since no remnants of aragonitic structures are observed. Further diagenetic processes that have been described for speleothems are*

*dissolution-precipitation under the influence of changing water chemistry of the drip water. The particular case at hand here, is diagenesis due to the contact with siliciclastic sediments and organic matter (bat droppings) that corrode the speleothems covered by these sediments. This is a diagenetic process that, to the best of our knowledge, has not been described in previous speleothem studies. From more general carbonate diagenesis literature, there are descriptions of microscale penetration of diagenetic dissolution-precipitation fronts that lead to neomorphic diagenesis that does not necessarily lead to changes in the crystal size.*

*We have rewritten the text of this section to contain the relevant information on the diagenetic processes, with added references.*

3.  I wonder how important are trace element results in this manuscript. It seems to me that those results are quite obvious and probably not too important (eg. avoid black layers when sampling is a quite usual procedure). Please, be more concrete about the utility of these proxies here.

*To our opinion, the trace element data are important, as they visualize diagenetic patterns that are relevant for the dating (the U concentrations are diagenetically enriched) and furthermore show how the extent of chemical diagenesis goes hand in hand with macroscopical indicators (discolorations, lamination changes, etc). The eventual utility for sampling speleothems lies in this coupling between the easily visible changes to the trace element chemistry. We feel that deserves to be described. We have rephrased some text, to make that clearer than it was.*

4.  I think that some interpretations that are based on only one date should be avoided or at least moderated. I am talking about climate inferences from growth cessation of only one sample, for example (lines 457, line 464, etc).

*This has been adapted*

5.  Detailed discussion about the final chronology and how it may change previous hypothesis would be needed as a kind of final conclusion of this manuscript.

*We have added a summarizing figure*

Some other minor comments:

- line 31: be more concrete about the "state-of-the-art" U-Th dating

*OK, done, in line with the extra info around the U-Th analysis progress discussed above.*

- line 90: you need to add the methods to prepare thin slides

*have added something about speleothem slab cutting and that thin sections were made off the slabs by cutting and polishing.*

- figure 3: numbers are too small. It may be better to split the stratigraphic column in two and make a squared-figure.

*There was some internal discussion about that, and we believe this depends on how this figure is placed in the paper. We will gladly increase the font size if the lay out of the ms requires that*

- section 2.2.: please, indicate which sample was dated in every lab.

*We have indicated this information in Table2: Results of the 230Th/U-dating of the Scladina samples.*

- section 2.3. can be added at the end of section 2.2

*This has been done*

- section 2.6: please indicate in which lab where the fluid inclusion analyses carried out

*This has been done.*

- section 2.8: this does not belong to materials and methods. Cave site and regional climate should be section 3.

*Has been done.*

- caption fig. 4. "shows clear discoloration towards the right side of the stalagmite". Is this important if you don't sample it?

*We think so, because the discoloration is one of the macroscopic indicators that suggests diagenesis. It is why we do not sample it.*

- section 3.2. too long and unclear. Please, consider to include all that information

about every speleothem in a table.

*We have adapted this section.*

● table 2: be sure that units are the same in Melbourne and Xian labs. (eg. activity versus atomic ratios).

*checked.*

● line 333: which would be the isotopic values you would expect if there was a "clear diagenetic imprint"? why? Please, explain.

*This section has been removed.*

line 434: the title is not correct to me. This is not a paleoclimate reconstruction but just few data on particular time windows. This has to be clear. These records do not overlap, they do not provide a continuous sequence.

*This issue was picked up by reviewer 2 as well. The title has been changed accordingly.*

● line 458 and beyond: talking about growth cessation and its association with climate deterioration with only one sample is no a robust argument. Please, moderate your interpretations about past climates.

*That section has been adapted*

● line 475: Sancho et al., (2015) can be also cited as evidences of a quite wet MIS6 in the Mediterranean region:

Sancho, C., Arenas, C., Vázquez-Urbez, M., Pardo, G., Lozano, M. V., Peña-Monné, J. L., Hellstrom, J., Ortiz, J. E., Osácar, M. C., Auqué, L., and Torres, T.: Climatic implications of the Quaternary fluvial tufa record in the NE Iberian Peninsula over the last 500 ka, Quaternary Research, 84, 398–414, https://doi.org/10.1016/j.yqres.2015.08.003, 2015.

*Reference added. Relevant indeed*

● line 490: please, indicate the uncertainty of the date. I doubt that its precission was so high to associate it to the transition from GI-21 to GS-20.

*This has been clarified. The date is at the very base of GS-20. The important part here is that the stalagmite grew during a glacial period, which was climatologically highly variable regarding D-O changes. Interestingly some more speleothems grew during this period in NW European caves and show that the climate during this period was mild*

*enough to permit speleothem growth.*

- lines 490-495: overinterpretation.

*We don't immediately see what is considered overinterpreted in this section? Is it the link to the SST's? The most important message here is that speleothem growth indeed occurred during a milder period (corresponding to GI 14) inside a glacial period and that it coincides with a period of higher SST in the marine record. Even though there are only two ages that correspond to GI14, of which one has rather high 232Th, we feel that this is relevant information. We rephrased slightly so that the uncertainties are more clearly voiced.*

- section 4.5 may be earlier to make clear what is new in this manuscript and how it has changed previous conclusions about the evolution of Neanderthas in Scladina cave.

*Now section 4.3 has been reduced and 4.4 removed, so that the emphasis is more on the dating aspect on the paper, this section has automatically become more central to the MS.*

- line 557: "A minority" is too optimistic, I am afraid.

*That has been rephrased to indicate that it was a significant part.*

*We thank reviewer 1 for the work and attention invested in our manuscript*

Reviewer 2: (our answers in Italic)

This paper presents some new U-Th dating results obtained on speleothems intercalated in a significant archaeological stratigraphy. The authors focus on the methods they used to scrutinize the speleothems and target the best sections in order to avoid the influence of diagenetic processes on the dating results. They obtain a new chronology, more complete and robust that the one previously available.

General comments: The description of the sampling, which is core to the rest of the discussion, is insufficient; some extra figures and tables should be provided to make all the information available.

*We have made some smaller clarifications at different sections in the text, and we have removed the oxygen isotope data of the speleothems from the ms entirely, we believe these issues are clearer now.*

The palaeoclimatological interpretations are beyond the scope of the paper, and rest on scarce and discontinuous data (dating mostly). They should be kept to a minimum by comparison with the benthic stack or the SST as in Fig. 6. Further, the section 4.4 discussing the box plots of stable isotope data is arguably the weakest part of the paper and should probably be removed. As a consequence, "palaeoclimatological data" should be withdrawn from the title.

*Title has been be amended (reviewer 1 suggested this too), section 4.4 removed, and 4.3 shortened so that the emphasis has shifted more to the dating and diagenesis.*

In order to support the significance of this work, it would be useful to provide a comparison of previous dates/age models with the one produced here, and to explain the most obvious consequences for the hypotheses of the archaeologists, especially regarding the age of the Neanderthal mandible.

*We have added a figure for this.*

Specific comments and technical corrections:

Fig. 1, Left: Font is too small
*is fixed.*

Fig 2 A: Not sure I understand this cross section. What is shown in brown? Sediment fill? How then is it possible to have bedrock less than a meter thick separating it from Sous Saint Paul cave? Has this lower cave been entirely filled? More textual description of the site would be useful here.

*We've clarified this in the text.*

Line 62: Typo Sous Saint Paul cave

*fixed*

Line 105: "2012#1&2 are two pieces of the same stalagmite in stratigraphic continuity with 2012#1". There is no "2012#1" listed in table 1.

*That sentence was a bit unclear, and is adapted.  We have removed Table 1.*

Section 2.2 U-series analyses: The dating was performed in 4 different labs and it seems, while reading this section, that the information provided on the methods are either incomplete, inconsistent or redundant. Hence this section would be greatly improved and be of greater interest to the reader if re-written, showing what is common between the different labs (overall, the chemistry is quite similar) and enhancing the differences (type of spike, standard, instrument?). The information about sample size and extraction is provided, briefly, only for those ran at MPIC; it is important, however, to document this information for all the samples (e.g. how? powder or prisms? dental drill or saw or corer or else? position on speleothems?).

*This section has been rephrased, following these suggestions*

Line 159: "corrected for detrital contamination assuming a 232Th/238U weight ratio of 3.8 for the detritus". Please cite a reference?

*Has been done.*

Line 171: Correct "The in-house CaCO3 standards VICS (...) was analyzed"

*done*

Line 178: "A single dripwater sample": please elaborate, e.g. if it's coming from a vial collecting monthly drip water or just the last hour or minutes of dripping.

*clarified.*

Line 186: "0.5-1.0 g chips of speleothem calcite" please elaborate on how these "chips" were detached from the speleothem, and in what specific locations (a figure would be useful).

*We explained that in the text, but prefer not to show it in a figure.*

Line 205: "Drip rates in the cave have been observed to be variable": is there any reference or data to support this? "This likely relates to the comparatively small hydrological catchment area": in comparison with what?

*both points have been clarified in the text: estimated dripwater catchment size is visible in fig 1*

Line 225: Phosphorus.

*OK, fixed*

Fig. 4: What are the holes on the slab? FI samples? If so, how come they were not extracted with a dentist drill in order to follow the laminae and optimize resolution?

*Holes on this slab were made for various analyses in student projects. The FI samples in this study were cut with a Dremel drill equipped with a rotary blade saw. We have added some text explaining that.*

Section 3.2: This section is hard to follow without the support of a figure showing the position of the samples or a summarizing table.

*We try to avoid adding too many figures to this manuscript, so we are hesitant to add photos of the stalagmites here. The sample positions (depth from top) are in Table 2, and we will further try to rephrase this section for clarity.*

*We have shortened and integrated this section with the earlier section on collected material. We believe this has improved the clarity of the text*

Fig. 6: Please indicate the color code for the U-Th dates.

*Thanks for spotting that, a legend has been added in the fig.*

Line 304: Grey bars (not green).

Line 308: Green bars (not grey).

*fixed.*

Line 333: "range of values does not directly indicate a clear diagenetic overprinting of the original signal." What do you mean? What values would you expect if there was diagenetic alteration?

*This section has been removed*

Line 335: Instead of "equilibrium conditions", this may reflect changes in precipitation conditions, influencing isotopic fractionation.

*This section has been removed*

Fig. 7: Not sure of the pertinence of this figure. What are the actual data encompassed in the box plot representing? We lack information about the sampling (missing in Methods).

*This figure has been removed, as suggested.*

Apparent contradiction in the reasoning with line 410: "supports limited or absent diagenetic alteration" and further, line 416: "diagenesis of speleothem calcite weakened fluid inclusion stability".

*Has been clarified. The fluid inclusion isotope section is now rephrased more in relation to the effects of diagenetic alteration*

Also, this part needs more details: Line 414 "rather resembles diffusive water loss from speleothem calcite as an analytical artefact, reported in a recent study by Fernandez et al. (2023)." And line 416: "facilitating diffusion of water out of the sample before analysis". What do you mean by "before"? Is it between the sampling and the crushing or is it in the cave setting? What kind of analytical artefact are you referring to?

*This is partial water loss that happens in the instrument, as the sample is brought to 110 degrees C for ~20 minutes before crushing and analysis. The Fernandez et al paper details this process. This has been clarified in the text.*

Section 4.3: "Palaeoclimatological backdrop": this part suffers from overinterpretation, or simplification, given the available data and their precision. I am not sure that the conclusion (i.e. that speleothems grow during periods of warmer climate) requires such a long presentation; the results of Fig. 6 are, in my opinion, sufficient to draw this conclusion.

*We have adapted this so that the emphasis is on stalagmite growth during warmer times, coupled to Fig 6.*

Lines 453-455: unclear.

*rephrased.*

Line 483: "Since some signs of diagenetic overprint have been observed in stalagmite 2010#6, we therefore suspect this particular age may be inaccurate." Then perhaps mention should be made of this in table 3 (FI results) and table 2 (e.g. add a column to warn about suspicious results or reversed ages).

*We have kept our concern around the validity of this age, on paleoclimatological grounds, in the paper, but describe better that it technically can not be discarded so easily, so that the option that this age is realistic remains the default.*

Section 4.4: This paragraph seems to me the most arguable and the least necessary

in this paper. As pointed out before (see comment about Fig. 7), we don't know how the data were obtained and what they actually represent, which makes the results difficult to discuss. Moreover, it's hard to comprehend why the isotopic values between a glacial and an interglacial would be less different than between the MIS 5e and the Holocene for example. It's also hard to believe that the MIS5e was one the driest interval, given the existing literature about LIG palaeoclimate in Europe. The seasonality might indeed be a factor but as you state it, your dataset does not allow you to resolve it. Overall, this part raises more questions than it solves and it does not add anything crucial to the core of your paper, hence I'd suggest to remove it.

*The section has been removed.*

Line 516: "Our data allow for several changes and additions to the existing age model". It would be useful and convincing to add a figure to show this, comparing the previous age model with the one you produced.

*We worked on the text, and added a summarizing figure.*

Line 536: "to assign ages to the stratigraphy of the underlying Sous-Saint-Paul Cave". One could argue that you provided a maximum age for the base of the SSP Cave stratigraphy but what do we know about the sedimentary processes and how this gallery was filled, in comparison with the one of Scladinia? Could they fill at the same time or were they necessary filled successively, starting with SSPC?

*It seems most likely that SSPC was filled first, given the ages we have now and the general superposition of SSPC and Scladina Cave that can be observed on-site. We have limited age data, of course, but that is more than before, when we had no radiometric age information at all for SSPC. The cave profile is  explained in text and figure caption in a bit more detail now.*

Conclusion: "Fluid inclusion isotope data of the speleothem samples predominantly plot close to the GMWL" and "fluid inclusion isotope data also provide a clear indication of the good preservation of most of the speleothem material selected, as fluid inclusions are considered to be very sensitive to diagenetic alteration. A minority of the fluid inclusion data trends away from the GMWL": This seems quite contradictory with what I understand from lines 412-416. Perhaps this latter paragraph should be clarified, or the conclusion should be more nuanced.

*We believe we have clarified this in the text now.*

Line 549: "One notable consequence of the new U-Th ages is that a juvenile Neandertal mandible, previously placed in Marine Isotope Stage 5a or 5b, could now potentially date back to Marine Isotope Stage 5d". Please explain why this is important, how it affects existing hypotheses.

*A follow-up archeological paper will detail the archaeological information and consequences for the Neanderthal story. Here, the purpose of the paper is to show and discuss the chronology of the stratigraphic sequence and shortly the consequences for the age of the mandibula. Indeed, more information can be deduced from this new chronology but this is outside of the scope of this paper.*

Fig A1. Please define the holes that are distinguishable on each slab; it would actually be useful to label each of them to make the information fully available: anyone should be able to see the exact position of any dating sample or FI sample on the slabs.

*The holes were made to take samples for several earlier student projects and do not necessarily coincide with analyses done in this study. We have add a line to clarify that. The "Depth from Top" position of all the samples in this study is indicated in Tables 2 and 3.*

Line 586: who is "A.J.v.N"?

*That is Marjan van Nunen. First name and initial are correct, but different from each other. We have changed this in the paper to avoid confusion.*

Data availability: they are not yet available in the Edmond database.

The Edmond database is activated for the trave element data, and the link added to the ms

References: Needs a careful check of formatting and homogenizing.

*Must be in right format now*